# Five-Year Experience of the Groupe de Recherche Action en Santé (GRAS) Clinical Laboratory, Burkina Faso, in Participating into an External Proficiency Testing (EPT) Programme

**DOI:** 10.3390/diagnostics16010036

**Published:** 2025-12-22

**Authors:** Amidou Diarra, Issa Nébié, Noëlie Béré Henry, Alphonse Ouédraogo, Amadou Tidiani Konaté, Alfred Bewentaoré Tiono, Sodiomon Bienvenu Sirima

**Affiliations:** Groupe de Recherche Action en Santé (GRAS), Ouagadougou 06 BP 10248, Burkina Faso; i.ouedraogo@gras.bf (I.N.); n.henry@gras.bf (N.B.H.); a.ouedraogo@gras.bf (A.O.); t.konate@gras.bf (A.T.K.); a.tiono@gras.bf (A.B.T.); s.sirima@gras.bf (S.B.S.)

**Keywords:** external proficiency testing, accuracy, performance, clinical laboratory, Burkina Faso

## Abstract

**Background**: The clinical research laboratory plays a pivotal role in the execution of clinical studies. The accurate and consistent registration of patients is dependent on the competent use of laboratory equipment and manual techniques by technicians, ensuring the reliability of the data collected. To support these activities, the Groupe de Recherche Action en Santé (GRAS) has been registered with the College of American Pathologists (CAP) and the Clinical Laboratories Services (CLS) in Johannesburg, South Africa, for external proficiency testing (EPT) of its laboratory, as part of our commitment to quality assurance. The following report details the performance achievements over the past five years. **Methods:** Proficiency testing (PT) samples are dispatched to GRAS Lab three times a year (quarterly) and the results are generally returned within two to three weeks. In the field of parasitology, challenge specimens were prepared as follows: thick and thin blood films were stained with Giemsa and mounted with strips to protect them for multiple uses. Photographs, also known as whole slide images (WSIs), were also taken. For the biochemistry and haematology tests, a set of five samples were received for processing. All evaluations were carried out in accordance with the GRAS laboratory’s internal procedures. **Results:** The CAP laboratory’s performance in terms of the diagnosis of malaria and other blood parasites from 2020 to 2024 was 97.3% accurate (ranging from 93.33% to 100%), with 93.33%, 100%, 100%, 93.33% and 100% achieved in 2020, 2021, 2022, 2023 and 2024, respectively. The number of microscopists evaluated annually has been subject to variation according to operational staff at the time of evaluation. A total of 31 microscopists were enrolled in the CLS PT scheme, of which 73.9% were classified as ‘experts’ and 19.2% as ‘reference’ microscopists. In the field of haematology, the PT demonstrated 100% accuracy over the four-year study period. This outcome is indicative of the high-performance levels exhibited by the automated systems under scrutiny and the comparable nature of the data produced by these systems. The same trend was observed in the biochemistry PT results, with an overall score of 92.12%, ranging from 78% to 100%. **Conclusions:** Proficiency testing has been shown to be an effective tool for quality assurance in laboratories, helping to ensure the accuracy of malaria and other blood parasite diagnoses made by microscopists, as well as the results generated by automated systems. It has been instrumental in assisting laboratories in identifying issues related to test design and performance.

## 1. Introduction

The accuracy of laboratory results is critical for clinicians to make decisions about treating patients or assessing the efficacy of drugs or vaccines in a clinical research setting.

This challenge is more pronounced in laboratories in countries with limited resources (LRLCs), where a combination of factors play a role, such as limited financial resources leading to inadequate infrastructure, equipment maintenance and training of competent personnel.

The Groupe de Recherche Action en Santé (GRAS) is a private research organisation with a vision to become a hub of excellence in biomedical research in West Africa. One of its main missions is to design and carry out clinical, operational and basic research studies to identify new tools for controlling diseases of interest, while monitoring the effectiveness of existing ones. To this end, GRAS has set up a technical platform to ensure highly efficient clinical laboratory practices. While accreditation bodies are not yet widespread or accessible to all institutions, external proficiency testing (EPT) may be a good alternative for clinical laboratories looking to build a high-quality system that performs satisfactorily [1,2].

Since 2020, GRAS has enrolled its laboratories in external quality control schemes with organisations that hold international recognition, in order to maintain high-quality standards. Thus, the clinical laboratory has been registered with the College of American Pathologists (CAP) for proficiency testing in haematology, clinical chemistry, and blood parasite surveys, while for malaria diagnosis by microscopy, the blood parasite proficiency testing scheme of the Clinical Laboratory Service/National Institute for Communicable Diseases (NICD) in South Africa was utilised.

This manuscript presents the results of five years of GRAS’s clinical laboratory’s continued participation in both PT schemes.

## 2. Materials and Methods

### 2.1. Description of the Clinical Laboratory of GRAS

The main laboratory is located in Ouagadougou, the capital city of Burkina Faso. Two secondary satellite research laboratories are situated at a distance of 100 km (the Health District of Sabou—The Nando region) and approximately 500 km (the Health District of Banfora—The Tannouyan region) from the capital city. The GRAS biomedical laboratory is structured around two main platforms. The first is a clinical biology platform, which investigates key laboratory tests relevant for clinical trials. These tests explore the safety of products (vaccines and drugs) under investigation and the medical care of study volunteers. The second is a fundamental biology research platform. The main goal of this platform is to gain a clear understanding of the pathogenesis of infectious diseases. This understanding is used to direct the development of new control tools. This laboratory provides essential support for clinical trials and research activities that involve the provision of medical care to patients. The instrument is equipped with standard, calibrated instruments for routine biomedical analysis, including biochemistry, haematology, bacteriology, immunology, parasitology, cellular immunology, gene sequencing and genotyping. The facility is equipped to conduct in-depth biological analysis on the pathogenesis of major infectious diseases, including emerging diseases, vectors and other intermediate hosts. Each instrument is subject to a preventative maintenance plan, carried out by a qualified technician on a minimum of an annual basis.

The laboratory is staffed by professionals with a range of scientific backgrounds, including biotechnologists, biologists and pharmacists. All personnel are regularly trained in Good Clinical Practices (GCP) as well as Good Clinical and Laboratory Practices (GCLP). In addition, in-house competency assessments are conducted on an annual basis.

Each biological test is carried out in accordance with internal controls for routine practices and research experiments.

The laboratory is often evaluated by sponsors prior to the launch of clinical trials. Staff members are retrained as and when necessary.

### 2.2. Architecture of the External Quality Control Schemes

Since 2020, GRAS has participated in two proficiency schemes. Firstly, it has joined the College of American Pathologists (Blood Pathogens, Biochemistry and Hematology) to assess the performance of laboratory quality systems. Secondly, it has collaborated with the Clinical Laboratory Services of South Africa to evaluate the proficiency of microscopists in malaria and other blood parasite diagnosis.

Each participating laboratory was required to register online in early December, via the CAP or CLS platforms, and pay the relevant fees, prior to receiving the proficiency testing specimens.

### 2.3. Profiency Testing with College of American Pathologists (CAP)

The College of American Pathologists’ (CAP) programme is one of the world’s largest external quality assessment programmes. As such, it provides an unparalleled selection of challenges and the largest existing database for interlaboratory comparison. Performance on CAP Surveys is not the sole indicator of a laboratory’s performance. Rather, PT indicators define what is required to assess, manage and improve quality [3].

PT samples are couriered to participating laboratories three times a year (quarterly) and the results are generally returned three weeks after the samples are shipped.

#### 2.3.1. Microscopy

In the field of microscopy, challenge specimens are thick and thin blood films (four or five per shipment) that have been stained with Giemsa and mounted with strips to protect them for multiple uses. Alternatively, photographs, also known as whole slide images (WSIs), can be used. WSIs have been proven to be an effective training tool in a variety of graduate education settings [4]. The target organisms are malaria parasites and other blood parasites. The procedure for preparing these slides also follows standard laboratory procedures, as outlined in the WHO manual and in accordance with the safety requirements of service providers [5]. At GRAS, the blood smears are assessed independently by each laboratory microscopist, and the results are collected and submitted by a third person, usually the head of the laboratory who is not involved in the slide reading process.

#### 2.3.2. Clinical Chemistry and Haematology

When conducting clinical studies on drugs or vaccines, it is vital to consider that biochemistry and haematology assessments are key to thoroughly ascertain the safety and tolerability of the investigational products under evaluation. Please refer to Table 1 for further details. Five samples are received and analysed for haematology and biochemistry assessment at each evaluation time point. The results from the analysis are submitted online via a dedicated platform. The samples are collected from three different study satellite sites (Banfora, Sabou and Ouagadougou) as part of the institutional internal control systems. The equipment used for biochemistry and haematology is, respectively, the INDIKO Plus 864 (Thermo Fisher Scientific Oy, Analysers & Automation, Clinical Diagnostics, Ratastie 2, Vantaa, FINLAND), SPIN 200E (Spinreact, SPAIN) and Mindray BS230 (Mindray, Shenzhen, CHINA) and Sysmex XN330 and Sysmex XN550 (Sysmex Corporation, Kobe, JAPAN). The laboratories process and analyse PT/EQA samples in the same way as routine clinical patient specimens based on their internal approved SOPs.

#### 2.3.3. Sample Evaluation and Scoring

Following submission for evaluation, the laboratory’s individual performance is graded using specified CLIA scoring criteria. These criteria are applied either by peer group or by an assigned target determined by selected reference laboratories.

Results obtained at different time points during the year-long evaluation were entered into an Excel worksheet to enable statistical computations and analysis. These results were then compared to estimate the accuracy and performance level of the participants or the laboratory over the years of the evaluation.

The College of American Pathologist PT evaluation (Parasitology) is based on sensitivity and specificity, as well as accuracy and precision for biochemistry and haematology.

A performance score of ‘satisfactory’ requires a minimum cut-off score of 80%.

For the CLS microscopy PT assessment, each laboratory technologist was assigned a grade based mainly on malaria parasite detection, identification and *P. falciparum* asexual parasite counting (see Table 2).

### 2.4. Profiency Testing with Clinical Laboratory Services (CLS)

This PT Scheme is produced by the Parasitology Reference Laboratory, NICD, NHLS, South Africa, and focuses on microscopy. The PT samples are couriered out to the reference laboratory three times a year, and results are generally received four weeks after shipping. The results of the study were compiled into individual reports and commentaries.

A total of 20 specimens were required for the evaluation. These were thick and thin blood films that had been stained with Giemsa. The thick blood films were mounted with strips to allow for multiple readings.

This scheme is designed to evaluate the performance of an individual microscopist, as opposed to the performance of the laboratory as a whole. It is therefore critical that microscopists work independently. Individual participant results were submitted to a designated non-participating colleague(s) who collated the responses for the laboratory. These were then submitted to CLS/NICD via a dedicated platform.

The accuracy of malaria species identification and parasite counting is the key performance indicator for microscopists. Parasite counts are scored as acceptable if they are within 25% of the true count (according to the validators used by CLS/NICD). The competence level achieved is based on whether the microscopist obtains the right results for parasite detection, parasite species identification and *P. falciparum* asexual parasite counting. For instance, to attain ‘Expert’ certification, a microscopist is required to achieve a minimum of 90% accuracy in parasite detection, speciation, and 50% of parasite counts must fall within a 25% margin of error compared to the true count (reference). Table 2 provides a summary of the competencies grading.

## 3. Results

During the five-year participation period, the parasitological laboratory was evaluated 12 times instead of 15. The three missed evaluations took place in 2022 (Q1 and Q2) and 2024 (Q1) due to the laboratory’s late registration in the CAP system. For the same reason, the haematology and biochemistry evaluations were missed in 2021, 2022 and 2024 in Q1 (first PT evaluation), respectively, for haematology and biochemistry. The summary of the number of PTs performed during the years of follow-up is presented in Table 3.

### 3.1. Parasitological Laboratory Accuracy Score from 2020 to 2024 of CAP PT Evaluation

The parasitological laboratory achieved a successful performance score of ≥80% during the five-year evaluation period, demonstrating its consistent accuracy. Notably, no unsatisfactory scores were recorded over the five-year PT participation period. *Plasmodium falciparum* (Pf) as outlined in Table 4, the performance of the Non-*Plasmodium falciparum* (Non-Pf) and other blood parasite detection tests are demonstrated. The CAP performance rate of the laboratory was 94.16%, ranging from 83.33% to 100%.

### 3.2. Haematology Laboratory Accuracy Score from 2021 to 2024 of CAP PT Evaluation

Table 1 outlines the primary parameters evaluated for haematology. Over the course of the four-year participation in the CAP PT evaluation, the score obtained during the first (Q1), second (Q2) and third (Q3) quadrimester evaluations were consistently 100% for all assessed parameters.

### 3.3. Biochemistry Laboratory Accuracy Score from 2021 to 2024 of CAP PT Evaluation

As outlined in Table 1, the primary parameters evaluated in biochemistry include triglyceride, uric acid, cortisol, thyroxine free (Th4, Free), human chorionic gonadotrophin (HCG), and thyroid-stimulating hormone (TSH). Over the course of the four-year participation in the CAP PT evaluation, the score for these primary parameters was consistently high, with the exception of total bilirubin (Q2, years 2021 and 2022), calcium (Q3, year 2023), total cholesterol, creatinine, glucose, and sodium (Q2, year 2022), albumin, chloride, creatinine, glucose (Q1, year 2023), and magnesium (Q2, year 2023). The overall score performance achieved a high percentage of 92.62%, ranging from 78% to 100% (Figure 1).

### 3.4. Parasite Identification and Counting Performance from 2020 to 2024 of CLS PT Evaluation

The CLS PT evaluation shares many similarities with the World Health Organization (WHO) External Competency Assessment for Malaria Microscopy (ECAMM). Both are based on the identification of malaria parasites and the counting of *P. falciparum* asexual parasites. However, there are some key differences. Firstly, 20 slides are used, as opposed to 5, as used in the CAP. Secondly, a higher score is required to qualify as a ‘reader’ (expert or reference). This additional requirement puts greater pressure on the expertise of the laboratory and the competence of the individual laboratory technologists. The number of laboratory technologists participating in this PT evaluation varied for each time point, depending mainly on workload. In total, 31 laboratory technologists participated in these CLS PTs, with 15 participating over the five-year evaluation period. The overall performance ranged from 94.15% to 97.48% for parasite detection and identification, and from 54.64% to 60.73% for parasite counting. The results are presented in Figure 2.

In addition to the primary focus on the identification and enumeration of malaria parasites, the CLS also evaluates participants’ individual performance in the detection and identification of other blood parasites such as Babesia, Microfilaria, Trypanosoma, Borrelia, Leishmania, etc. The combination of parasite detection, identification and *P. falciparum* asexual parasite counting is used to grade participants, as illustrated in Figure 3.

In addition, we evaluated the proportion of microscopists who transitioned from qualified readers to non-qualified or vice versa during the evaluation period. A total of 12.67% of participants initially classified as qualified readers changed their status to become non-qualified during at least one evaluation time point. Please refer to Figure 4 for further details.

## 4. Discussion

In a clinical research laboratory, the accuracy of data is a key factor in clinical decision-making. Participation in an external quality assessment, such as proficiency testing (PT), is an essential requirement for accreditation and ISO 15189:2022 (International Organization of Standardization (ISO) 15189, Annals of laboratory Medicine, volume 37, number 5, USA 2017) for medical laboratories [6]. Nevertheless, producing accurate and reliable results through PT is challenging for clinical laboratories in developing countries. The main challenge lies in the inadequate or inefficient allocation of resources to laboratory services [7]. It is fortunate that a paradigm shift has occurred in the last 15 years, driven by the investment of governments and private health sectors in the three main pathway programmes for implementing quality management systems (QMS). These programmes were launched in 2009 by WHOAFRO and include Strengthening Laboratory Management Towards Accreditation (SLMTA), Stepwise Laboratory Improvement Process Towards Accreditation (SLIPTA) and Laboratory Quality Stepwise Implementation (LQSI) [8].

This stimulating ecosystem was an encouragement for the GRAS to embark into these external PT programmes during which key experiences were gained.

### 4.1. Why Participate in a Proficiency Testing Programme?

Ensuring the accuracy of laboratory results is of the utmost importance in all areas of health system operations. In clinical research, clinical decisions made by physicians for patient care are largely dependent on the findings from clinical laboratories [9]. Indeed, the involvement of volunteers in clinical studies (drug or vaccine assessment) is primarily contingent on laboratory results that provide the green light for safety and subsequent follow-up of the volunteer. Any error in a laboratory result has the potential to impact all study procedures and compromise the safety of the participants. The establishment of a robust quality system is imperative and warrants particular attention. The initial step in ensuring the reproducibility and accuracy of results utilised for patient management in any clinical laboratory is the implementation of standardised procedures. However, the WHO advises the utilisation of external agencies for the objective evaluation of a laboratory’s performance [10].

### 4.2. Is There Any Pre-Requisite for a Lab to Participate in a PT Programme?

Engagement of a clinical laboratory into a PT scheme requires a basic and fundamental organisation, based mainly on the 12 quality system essentials (QSEs) delineated by the Clinical and Laboratory Standards Institute (CLSI) [11]. Out of these QSEs, it is vital to implement a robust quality management system (QMS). A laboratory quality management system is a systematic, integrated set of activities that establish and control work processes from the pre-analytical through to the post-analytical stages. It also manages resources, conducts evaluations and makes continual improvements to ensure consistent quality results. Approaches to quality management in clinical microbiology were described previously by Bartlett et al. in 1967 [12]. Since 2020, the Groupe de Recherche Action en Santé (GRAS) has maintained a quality management system (QMS) that covers almost all aspects of the institution, including laboratory activities. Each laboratory activity is guided by a written, approved Standard Operating Procedure (SOP). The maintenance of the equipment is carried out in accordance with either manufacturer recommendations or internal experience of its use. Laboratory technologists undergo regular training on Good Clinical Practice (GCLP) and all SOPs prior to operating laboratory activities.

### 4.3. How to Sustain the Engagement and Good Skills in Diagnosis of the Lab Staff?

Maintaining the motivation of clinical laboratory staff requires effective management through training and effective organisation. A leadership and integrated institutional quality management system is essential, although the feasibility of these measures is highly dependent on financial resources allocated [13]. In order to overcome the challenges identified, the programme places significant emphasis on the continuous education of the laboratory staff, taking advantage of any emerging opportunity.

With regard to the microscopical PT challenge, it is acknowledged that the drawback of blood smear microscopy for the diagnosis of malaria and other blood parasites is that it is operator-dependent and requires initial and continuous training to maintain a high standard of testing. Such quality assurance practices are often difficult to implement in resource-poor countries [14]. Microscopy performed on stained films of peripheral blood is an essential reference standard for clinical trials of drugs, vaccines and diagnostic tests for malaria. This is because it can be used to detect, identify and quantify malaria parasites [15]. During the five-year participation in the CAP PT, the parasitological laboratory achieved a consistently high performance, with a 76.66% detection rate for other blood parasites and 100% for malaria parasites speciation and negative slide detection. The laboratory demonstrated a 100% sensitivity and specificity for malaria parasites assessment over the five-year period. This exceptional performance can be attributed to the rigorous internal quality management plan implemented in the laboratory several years ago. Continuous refresher training for the laboratory staff and strict adherence to operational procedures are crucial elements of this programme. A study conducted by Mary and colleagues from 2015 to 2019 examined the impact of a malaria diagnostic refresher training programme on the competencies and skills of medical laboratory professionals in malaria diagnosis. The study found that participants’ competency levels and skills in malaria microscopy (detection, speciation, and malaria parasite quantification) increased by approximately twofold after the training compared to the no-training scenario [16]. The training programme saw a marked improvement in the ability of medical laboratory professionals to correctly detect malaria parasites, with a median score rising from 64% to 87%. Similarly, the competencies of medical laboratory scientists to accurately identify and quantify malaria parasites increased significantly, from a median score of 17% and 20% pre-test to 78% and 50% post-test, respectively. The implementation of Good Clinical Laboratory Practices (GCLP) led to a notable increase in the success rate of PT surveys, rising from 58% in 2009 to 88% in 2010. Additionally, a study conducted in Ghana by Ibrahim et al. in 2012 revealed a 35% decrease in error rate on PT [17].

At the individual level, the CLS PT evaluation on malaria and other blood parasites has shown a higher score of performance over the five-year evaluation period. It has a high ability for parasite detection, identification and counting. The level of expertise that ‘qualifies technologists as readers’ (Experts or Reference) ranges from 83.33% to 100% during the five-year evaluation period. This level of expertise is in line with the WHO standard, where any treatment of a malaria case should be based on evidence of the presence of the parasite in the patient’s blood and on a drug protocol assessment [18].

During the five-year evaluation of the CLS, the capacity to detect and identify malaria parasites was consistently 100%. The counting process was the primary concern, impacting the grading level of laboratory technologist. However, the means of parasite counting consistently exceeded 50%, aligning with the WHO External Competence Assessment (ECA) guidelines [19]. In the context of multi-parasitic infection, a range of parasites, including filariasis, leishmaniasis, trypanosomiasis, and Babesia, as well as bacterial and fungal species, can be present alongside malaria parasites [20,21,22]. It is imperative to have highly skilled laboratory technologists who are able to accurately identify and distinguish these parasites [23,24]. A high number of qualified laboratory technologists serving as readers across the study sites has been maintained for the benefit of the patients. This was highly appreciated during the monitoring process by the peer reviewers. The increased number of ‘non-readers’ (competent or in-training) shown in Figure 4 (from 8% to 16.6%) is mainly due to the new staff recruited during the years of evaluation. As we are aware, acquiring the skills required for malaria parasites diagnosis takes time and effort during training [15].

The involvement of GRAS Clinical Laboratory in EPT has been a resounding success in terms of establishing and refining methodologies, as well as the processing of study volunteer samples. The 100% accuracy achieved over the course of the four-year participation of the haematology laboratory is a testament to the high-performance capabilities of the automated systems, the consistency of data across these systems, and the robustness of the internal SOP employed. This has led to the generation of dependable results, directly benefiting study volunteers and the clinicians responsible for patient care. Notably, a study conducted by Chaudhry et al. in 2023 clearly demonstrated that 70–75% of medical diagnoses are made based on clinical laboratory reports, underscoring the direct impact of laboratory service quality on healthcare quality [9]. When assessing the safety of drugs or vaccines, biochemistry parameters are a crucial component of the evaluation. The direct impact of investigational products on vital organs, such as the liver, kidneys, electrolytes, heart, and blood gas levels, can be significant. During the PT evaluations, particular attention was given to these parameters. As illustrated in Figure 1, the overall biochemistry PT score performance was 92.62%, ranging from 78% to 100%, despite challenges encountered with certain parameters, including bilirubin, calcium, total cholesterol, creatinine, glucose, sodium, albumin, chloride, creatinine, and magnesium in 2022 and 2023. Internal investigations revealed that these unsatisfactory results were primarily due to technical issues and the non-specificity of PT tests to the various biochemistry automated systems utilised in our Clinical Laboratory.

In a study conducted by Tingting Li et al. in 2019, the distribution of PT test failure types was as follows: Clerical, 10.2%; Methodological, 16.2%; Equipment, 23.6%; Technical, 37.4%; PT Evaluation, 0.5%; Unexplained, 11.5%; and Other, 0.7%. Reasons unique to the PT process accounted for 15.6% of all reasons [25].

Another challenge that needs to be overcome if we are to successfully establish an EQA programme is the allocation of sufficient budgets and resources [26]. During the five-year participation period, the PT fees were consistently covered by the institution’s financial accountability service, which was advantageous. All other potential drawbacks that could have impacted the PT’s outcome, such as laboratory equipment and consumables, EQA sample management, logistics, and supply chain management of EQA samples, were integrated into a robust internal quality management system (IQMS).

## 5. Conclusions

Proficiency testing has been demonstrated to be an effective laboratory quality assurance tool. It has been shown to improve the skills of lab technologists in diagnosing malaria parasites and other blood parasites, as well as the quality of results generated by automates during clinical studies. It has also helped laboratories identify issues related to test design and performance. Furthermore, the ability to compare laboratory performance with others using the same or different methods on identical samples has highlighted issues related to test methodology or interpretation. This has helped in developing the best practice guidelines and standard policy. The laboratory was able to sustain high performance over the five-year evaluation period. The results of the PT have reinforced confidence in the results generated during clinical research and patient management, and thus in the credibility of the partners and the research community regarding GRAS research findings.

## Figures and Tables

**Figure 1 diagnostics-16-00036-f001:**
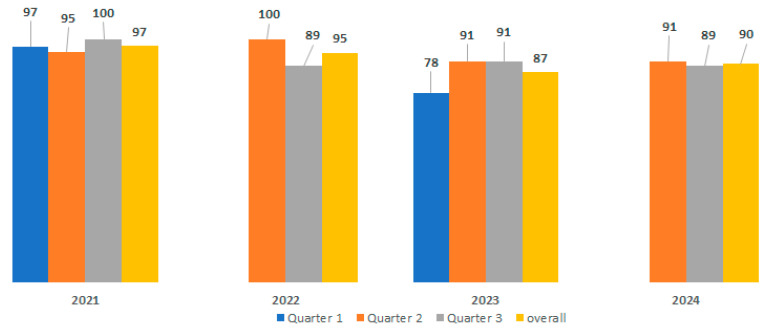
Overall biochemistry lab performance during the 4-year PT evaluation by the CAP.

**Figure 2 diagnostics-16-00036-f002:**
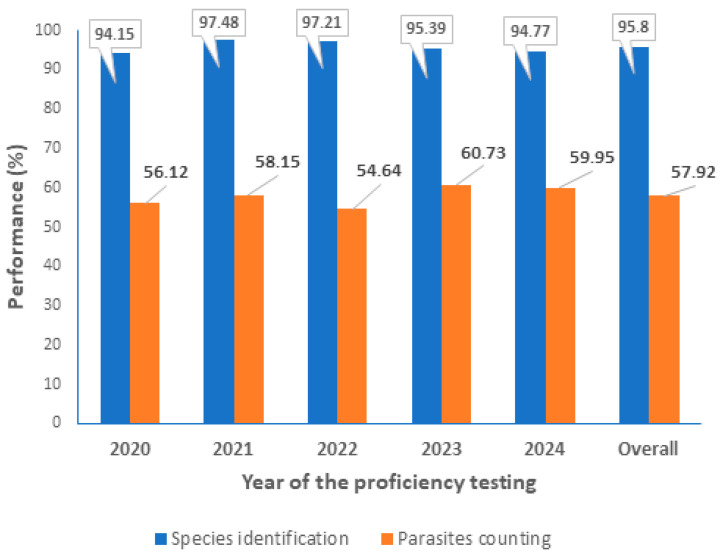
Parasite identification and counting performance during the 5-year evaluation by the CLS.

**Figure 3 diagnostics-16-00036-f003:**
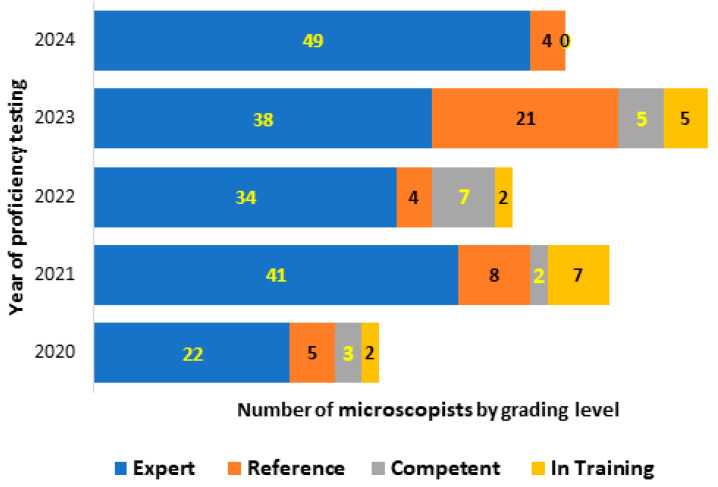
Participants grading profile during the 5-year evaluation by the CLS.

**Figure 4 diagnostics-16-00036-f004:**
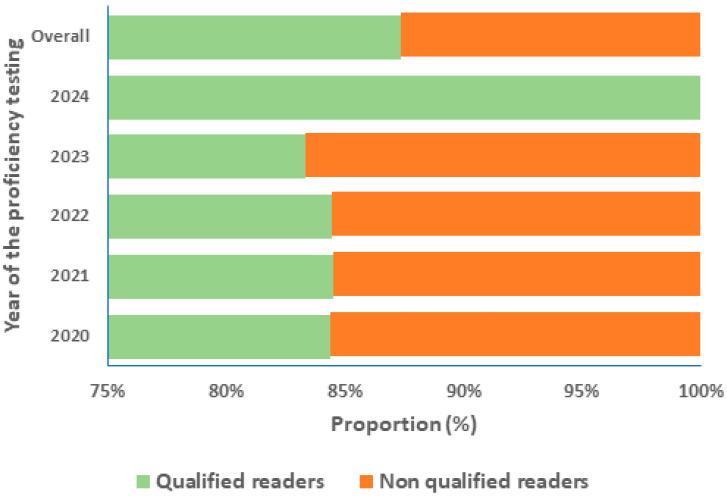
Qualified and non-qualified readers performance during the 5 year-PT evaluation by the CLS.

**Table 1 diagnostics-16-00036-t001:** CAP PT haematology and biochemistry parameters.

Haematology Parameters	Biochemistry Parameters	
Cell ID/Flow differential	Alanine aminotransferase (ALT)	Creatinine
Erythrocyte count (RBC)	Alkaline Phosphatase	Glucose
Haematocrit	Amylase	Lactate Dehydrogenase (LDH)
Haemoglobin	Aspartate aminotransferase (AST)	Potassium
Leucocyte count (WBC)	Bilirubin, Total	Sodium
Platelet count	Chloride	Urea Nitrogen
	Cholesterol, Total	

**Table 2 diagnostics-16-00036-t002:** CLS PT grading scheme.

Competency Level	Parasite Detection Performance (%)	Species Identification Performance (%)	Parasite Count Performance Within 25% of True Count (%)
Expert	90–100	90–100	50–100
Reference	80–89	80–89	40–49
Competent	70–79	70–79	30–39
In-Training	0–69	0–69	0–29

**Table 3 diagnostics-16-00036-t003:** Number of surveys carried out during the follow-up periods.

PT Parameters		Years of Evaluation					
		2020	2021	2022	2023	2024	Total
CAP	Haematology	ND *	02	02	03	02	09
	Biochemistry and Electrolytes	ND *	03	02	03	02	10
	Blood parasites	03	03	01	03	02	12
CLS	Blood parasites	03	03	03	03	03	15

* ND = not done.

**Table 4 diagnostics-16-00036-t004:** Parasite detection and counting performance during the follow-up periods.

Year	Pf Detection Performance (%)	Non-Pf Detection Performance (%)	Other Blood Parasite Detection Performance (%)	Counting Performance (%)	Overall Detection Performance (%)
2020	100	100	33.33	100	83.33
2021	100	100	100	100	100
2022	100	100	100	100	100
2023	100	100	50	100	87.5
2024	100	100	100	100	100
Overall (%)	100	100	76.66	100	94.16

## Data Availability

The original contributions presented in this study are included in the article. Further inquiries can be directed to the corresponding author.

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
