# Peer review of "Five-Year Experience of the Groupe de Recherche Action en Santé (GRAS) Clinical Laboratory, Burkina Faso, in Participating into an External Proficiency Testing (EPT) Programme"

_diagnostics, 2025, doi:10.3390/diagnostics16010036_

Round 1
Reviewer 1 Report
Comments and Suggestions for Authors
This manuscript is well written, but there are some points in the text that must be corrected:
At line 16, there is no description of what is PT abbreviation, as I have found for College of American Pathologists. although at the end o f the manuscript is the meaning of the abbreviation used, it is better to standardize the text;
at line 148, 207 and 221, Plasmodium falciparum has to be in italic;
At line 189, the writting laboratory has to be corrected.
Just as a suggestion, you could use an statistics parameter to improve the robustness of this study as the ANOVA test to compare the performance between technicians; the Standard Deviation and the Coefficient of Variation in order to measure the overall Precision and Consistency and also, analyze the Trends Over the Time, using i.e. Levey-Jennings control charts.
I think you have the data to improve this manuscript, but as it is, in my opinion, it can be sent for publication.
Author Response
“Lessons learnt from a Five-year experience of the Groupe de Recherche Action en Santé (GRAS) Clinical Laboratory, Burkina Faso, in Participating to an External Proficiency Testing (EPT) Programme “
Dear Reviewers,
Thank you for reviewing the manuscript MS: diagnostics-3869562 entitled: “Lessons learnt from a Five-year experience of the Groupe de Recherche Action en Santé (GRAS) Clinical Laboratory, Burkina Faso, in Participating to an External Proficiency Testing (EPT) Programme”.
Your comments and suggestions have been addressed as indicated below and highlight with blue colour. We hope that amendments made will meet your requirements.
Thanks so much for your time dedicated for the review
Best wishes
Response to reviewers’ queries
Reviewer's report:
Comments and Suggestions for Authors
This manuscript is well written, but there are some points in the text that must be corrected :
At line 16, there is no description of what is PT abbreviation, as I have found for College of American Pathologists. although at the end of the manuscript is the meaning of the abbreviation used, it is better to standardize the text;
Answer and comments:
A description of PT abbreviation has been done throughout all the manuscript when necessary
at line 148, 207 and 221, Plasmodium falciparum has to be in italic;
Plasmodium falciparum has been corrected in italic
At line 189, the writting laboratory has to be corrected.
Writing laboratory has been corrected at line 189
Just as a suggestion, you could use an statistics parameter to improve the robustness of this study as the ANOVA test to compare the performance between technicians; the Standard Deviation and the Coefficient of Variation in order to measure the overall Precision and Consistency and also, analyze the Trends Over the Time, using i.e. Levey-Jennings control charts.
Thank so much for the good suggestion. In fact, comparison of the performance between technicians was not an objective we wanted to access. The main concern was the number of “readers” at any time point of the evaluation to assure that we have qualified laboratory technologist to handle the lab activities.
I think you have the data to improuve this manuscript, but as it is, in my opinion, it can be sent for publication.
Thank you for your time dedicated for reviewing the manuscript
Reviewer 2 Report
Comments and Suggestions for Authors
I reviewed Lessons learnt from a Five-year experience of the Groupe de Recherche Action en Santé (GRAS) Clinical Laboratory, Burkina Faso, in Participating to an External Proficiency Test (EPT) Programme an interesting work. Any measure that improves the quality of laboratory results must be respected, but
- what are lessons -from title? I am not sure if the title matches the content of the paper.
- you must improve statisticals
- you must highlight the content of the pictures
- how you can evaluate External Proficiency Test (EPT) Programme with missing evaluations in biochemstry, haematology?
Author Response
“Lessons learnt from a Five-year experience of the Groupe de Recherche Action en Santé (GRAS) Clinical Laboratory, Burkina Faso, in Participating to an External Proficiency Testing (EPT) Programme “
Dear Reviewers,
Thank you for reviewing the manuscript MS: diagnostics-3869562 entitled: “Lessons learnt from a Five-year experience of the Groupe de Recherche Action en Santé (GRAS) Clinical Laboratory, Burkina Faso, in Participating to an External Proficiency Testing (EPT) Programme”.
Your comments and suggestions have been addressed as indicated below and highlight with blue colour. We hope that amendments made will meet your requirements.
Thanks so much for your time dedicated for the review
Best wishes
Response to reviewers’ queries
Reviewer's report:
I reviewed Lessons learnt from a Five-year experience of the Groupe de Recherche Action en Santé (GRAS) Clinical Laboratory, Burkina Faso, in Participating to an External Proficiency Test (EPT) Programme an interesting work. Any measure that improves the quality of laboratory results must be respected, but
- what are lessons -from title? I am not sure if the title matches the content of the paper.
Yes, we agree that the manuscript Title may not match with the content and a new Title was addressed as follow: “Five-years’ experience of the Groupe de Recherche Action en Santé (GRAS) Clinical Laboratory, Burkina Faso, in Participating to an External Proficiency Testing (EPT) Programs”.
- you must improve statisticals
ok
- you must highlight the content of the pictures
ok
- how you can evaluate External Proficiency Test (EPT) Programme with missing evaluations in biochemstry, haematology?
What we are trying to highlight here is just the missing time point of the evaluation as explained in the methodology section. Otherwise, evaluated parameters is the choice of the laboratory depending of the objective he wants to achieve.
Thank you for your time dedicated for reviewing the manuscript
technologists were systematically excluded from the pool or readers and undergone refresh training and evaluation to prepare them for the next evaluation.
For biochemistry and hematology, “bad EQA” results were also reviewed applying internal QC rules likes Westgard Levey-Jennings control charts to define the type of error and apply the corresponding correction.
Thank you for your time dedicated for reviewing the manuscript.
Reviewer 3 Report
Comments and Suggestions for Authors
I read the manuscript, dealing with a report of EQA results from a central laboratory and two satellite laboratories in Burkina Faso with great interest and want to congratulate the authors with their care for quality, and good EQA results.
Unfortunately, the English of the manuscript needs to be improved considerably. I consider asking a native English speaker to improve the language of the manuscript.
Besides that, I also have some punctual remarks for improving the manuscript;
1. Introduction, line 45-46: the authors menion 'lack of limited financial resources'. I guess that they want to write 'lack of financial resources', or 'limited financial resources'. Please correct.
2. section 2.2, line 102: the authors state that the CAP is the largest external quality assessment program in the world. Please give proof. Randox also claims to be the largest EQA in the world.
3. Section 2.2.3, line 141: the authors write that the PT evaluation of the CAP for parasitology, hematology and biochemistry is based on sensitivity and specificity. For parasitology, I can understand that. Is this also true for hematology and biochemistry ?
4. secton 2.3, line 152. The authors write that the samples are sent ot the reference laboratory. What do the authors mean here with the term 'reference laboratory' ?
5. Section 2.3, line 168: what do the authors mean with 'speciation' ?
6. Section 3, lines 174-180: mentionning with PTs the laboratory could not participate in, should not be mentioned. I recommend the authors to keep it short, and mentionning only that the laboratory could not participate in all the rounds. For the same reason, table 3 is superfluous.
7. General remark: I am missing details on how bad EQA results were handled and how the lessons learned from bad EQA results has improved the overall quality of the laboratory. Can the authors give more examples on this ?
Author Response
“Lessons learnt from a Five-year experience of the Groupe de Recherche Action en Santé (GRAS) Clinical Laboratory, Burkina Faso, in Participating to an External Proficiency Testing (EPT) Programme “
Dear Reviewers,
Thank you for reviewing the manuscript MS: diagnostics-3869562 entitled: “Lessons learnt from a Five-year experience of the Groupe de Recherche Action en Santé (GRAS) Clinical Laboratory, Burkina Faso, in Participating to an External Proficiency Testing (EPT) Programme”.
Your comments and suggestions have been addressed as indicated below and highlight with blue colour. We hope that amendments made will meet your requirements.
Thanks so much for your time dedicated for the review
Best wishes
Response to reviewers’ queries
Reviewer's report:
Read the manuscript, dealing with a report of EQA results from a central laboratory and two satellite laboratories in Burkina Faso with great interest and want to congratulate the authors with their care for quality, and good EQA results.
Unfortunately, the English of the manuscript needs to be improved considerably. I consider asking a native English speaker to improve the language of the manuscript.
Thank: The all manuscript has been reviewed by a native English speaker
Besides that, I also have some punctual remarks for improving the manuscript ;
- Introduction, line 45-46: the authors menion 'lack of limited financial resources'. I guess that they want to write 'lack of financial resources', or 'limited financial resources'. Please correct
Yes, the sentence has been corrected: “limited financial resources”
- section 2.2, line 102: the authors state that the CAP is the largest external quality assessment program in the world. Please give proof. Randox also claims to be the largest EQA in the world.
Yes, the sentence has been corrected: “one of the largest external quality assessment programs in the world”
Section 2.2.3, line 141 : the authors write that the PT evaluation of the CAP for parasitology, hematology and biochemistry is based on sensitivity and specificity. For parasitology, I can understand that. Is this also true for hematology and biochemistry ?
No! the sentence has been corrected: « PT evaluation (Parasitology) is based on sensitivity and specificity and on accuracy, and precision for Biochemistry and hematology »
- secton 2.3, line 152. The authors write that the samples are sent ot the reference laboratory. What do the authors mean here with the term 'reference laboratory' ?
The “reference laboratory” is the lab (referee) used by the CLS/CAP to evaluate others labs.
- Section 2.3, line 168: what do the authors mean with 'speciation' ?
“speciation” is the skill to differentiate malaria or any parasite species. Identification is also often used, but for better understanding, we use speciation.
- Section 3, lines 174-180: mentionning with PTs the laboratory could not participate in, should not be mentioned. I recommend the authors to keep it short, and mentionning only that the laboratory could not participate in all the rounds. For the same reason, table 3 is superfluous.
Thank for the suggestion
- General remark: I am missing details on how bad EQA results were handled and how the lessons learned from bad EQA results has improved the overall quality of the laboratory. Can the authors give more examples on this ?
Handled of “bad EQA” results: “bad EQA” results were handled applying mainly internal QC procedures and also PT program recommendations. For example, for malaria and others blood parasites assessment; the failed laboratory technologists were systematically excluded from the pool or readers and undergone refresh training and evaluation to prepare them for the next evaluation.
For biochemistry and hematology, “bad EQA” results were also reviewed applying internal QC rules likes Westgard Levey-Jennings control charts to define the type of error and apply the corresponding correction.
Thank you for your time dedicated for reviewing the manuscript.
Round 2
Reviewer 2 Report
Comments and Suggestions for Authors
Congratulations for your work, you sent an improved version.
Author Response
Manuscript ID: diagnostics-3869562
“Lessons learnt from a Five-year experience of the Groupe de Recherche Action en Santé (GRAS) Clinical Laboratory, Burkina Faso, in Participating to an External Proficiency Testing (EPT) Programme “
Dear Reviewers,
Thank you for reviewing the manuscript MS: diagnostics-3869562 entitled: “Lessons learnt from a Five-year experience of the Groupe de Recherche Action en Santé (GRAS) Clinical Laboratory, Burkina Faso, in Participating to an External Proficiency Testing (EPT) Programme”.
Your comments and suggestions have been addressed as indicated below and highlight with blue colour. We hope that amendments made will meet your requirements.
Thanks so much for your time dedicated for the review
Best wishes
Response to reviewers’ queries
Reviewer number: 2 Round 2
Reviewer's report:
I reviewed Lessons learnt from a Five-year experience of the Groupe de Recherche Action en Santé (GRAS) Clinical Laboratory, Burkina Faso, in Participating to an External Proficiency Test (EPT) Programme an interesting work. Any measure that improves the quality of laboratory results must be respected, but
- The English could be improved to more clearly express the research
Manuscript has been reviewed by a native English speaker
- Figures and tables must be improved.
Figures and tables were update
Thank you for your time dedicated for reviewing the manuscript